# Adaptation to Virtual Assessment during the COVID-19 Pandemic: Clinical Case Presentation Examination

**DOI:** 10.3390/dj11020045

**Published:** 2023-02-09

**Authors:** James Donn, J. Alun Scott, Vivian Binnie, Kurt Naudi, Colin Forbes, Aileen Bell

**Affiliations:** 1Department of Restorative Dentistry, University of Glasgow, School of Medicine Dentistry and Nursing, Glasgow Dental Hospital and School, Glasgow G2 3JZ, UK; 2Department of Dental Public Health, Dentistry University of Glasgow, School of Medicine Dentistry and Nursing, Glasgow Dental Hospital and School, Glasgow G2 3JZ, UK; 3Department of Oral Surgery, Dentistry University of Glasgow, School of Medicine Dentistry and Nursing, Glasgow Dental Hospital and School, Glasgow G2 3JZ, UK

**Keywords:** COVID-19, clinical assessment, dental Students, case presentation, online, dental education, prevention: SARS-CoV

## Abstract

Background: Case presentation assessment is common in both medicine and dentistry and is known under various names depending on the country and institution. It relates mainly to aspects of diagnosis and treatment planning and is considered highly authentic and useful. The COVID-19 pandemic necessitated the movement of this assessment from face-to-face to online. The aim of this investigation was to explore the students’ impressions of the two different examination modalities. With this information, a decision on future diets of this examination can be made to accommodate the students’ perspectives. Methods: Quantitative and qualitative data were gathered using an online, self-administered survey. Results: The students were split 50/50 regarding which assessment modality they preferred. Overall, they considered the online examination to be fair, and the majority agreed that the online format allowed them to display their knowledge as well as face-to-face. Conclusions: The delivery of case presentation examination is possible online. An online case presentation is a fair, useful, and authentic assessment that is appropriate to the needs of the faculty and students. Satisfaction with the two possible methods of conducting this assessment suggests it would be reasonable to conduct this examination online in the future.

## 1. Introduction

To ensure the competence of new graduates, it is essential that the assessment of final-year dental students encompass a range of modalities. Typically, assessments are conducted through written examinations, such as multiple choice, multiple short answers, and essay questions, which predominately assess the student’s knowledge, while objective structured clinical examination (OSCE) and case presentation (CP) are more applicable to clinical skills [1,2,3]. While OSCE can provide an overview of students’ clinical understanding, CP is often more specific and relates mainly to aspects of diagnosis and treatment planning.

CP assessment is common to both medicine and dentistry and is known under various names depending on the country and institution: ‘long case’, ‘oral case presentation’, and ‘case-based discussion’ [1,2,3,4,5,6]. While such evaluations are considered highly authentic [2,4] and useful [7], there is a discussion surrounding the reliability of this method [2] with a lack of standardization of questioning being inevitable where different patients are presented by individual students [8].

Case presentation is an integral part of final examinations, at our institution, where in conjunction with Multiple Short Answer (MSA) and OSCE, students are assessed over an extensive range of intended learning outcomes (ILOs). These ILOs are mapped to the UK General Dental Councils’ ‘Preparing for Practice’ document [9] and are essential in ensuring attainment at an appropriate level for graduation. The students are required to prepare their CP on a patient requiring treatment over three or more dental disciplines. This is predominantly a diagnosis and treatment planning exercise in our institution. All aspects of assessment, treatment planning, and delivery are carried out under supervision. The student collates information about the patient and their proposed treatment. This information is reproduced in poster form allowing the student to present their case orally. A formative ‘mock’ presentation is held halfway through the year with the summative ‘final’ examination held at the end of the academic year. Both examinations are 15 min orals assessed by two staff members, one restorative and one oral surgery/oral medicine, to ensure a breadth of evaluation. External examiners are also present at the summative assessment. Staff calibration is carried out for the summative examination.

The academic term 2020–2021 was extremely challenging due to the COVID-19 pandemic. While written assessments could be moved online easily, clinical examinations proved to be extremely challenging. Several institutions, including our own, succeeded in moving OSCE online: generally, to good effect [10,11,12,13]. To allow progression, it was essential that our CP examination also took place during this time. Using the same online approach developed for the virtual OSCE (VOSCE) [11], it was possible to accomplish this essential part of our assessment process.

The ‘Mock’ examination took place as a live face-to-face oral examination pre-pandemic, but the ‘Final’ examination was post-pandemic and had to be conducted online. The platform chosen was Zoom [14] as this was the most familiar platform for our staff and undergraduates. This examination was also conducted in real-time with the student screen facing at all times. A criticism of online examinations, in general, is the opportunity for students to access other resources during the examination process. This would not be possible in the context of this assessment where the student is visible to the examiners at all times.

The aim of this study was to explore the students’ impressions of the two different examination modalities. With this information, a decision on future diets of this examination can be made to accommodate the students’ perspectives. Student satisfaction with assessment is rightly one of the key themes of subject review within our university and, additionally, is a component of the annual National Student Survey by which universities are ranked within the UK.

While the initial driver towards online assessment was the Covid pandemic, it is not definite that there will be a return to face-to-face CP examination as the public health emergency recedes. Furthermore, the online option provides a future-proofed alternative should any future emergency situation arise.

## 2. Materials and Methods

### 2.1. Formative Case Presentation Procedure, Pre-COVID

After patient examination and treatment planning, the student collated information regarding their patient’s treatment. Clinical notes, photographs, radiographs, periodontal charting, study models, and results of any other special tests were all collected by students, where appropriate. From these, a poster summarising the patient data and treatment plan was produced. At the oral case presentation examination, the student presented the poster along with hard copies of all the supplemental information to two examiners in a 15 min, live face-to-face oral examination.

### 2.2. Summative Case Presentation Procedure, during COVID

The poster and all supporting clinical evidence were digitised and transferred to a ‘PowerPoint’ [15] presentation by the student. Students were assigned to individual breakout rooms within the Zoom [14] platform where they presented their cases to two examiners in a 15 min online examination. Rather than the examiner being able to handle any artifacts, they could move between slides of this virtual poster, showing clinical photos, radiographs, periodontal screening, etc. The student was always visible on the screen during the assessment process. The external examiners were present at this assessment and were able to move between the breakout rooms throughout the process.

### 2.3. Data Collection

Ethical approval was obtained from the University of Glasgow ethics committee.

The study involved administering a specially designed questionnaire to the dental cohort involved. The questionnaire was constructed using the Microsoft forms software. (See Appendix A).

This consisted of 9 items designed to assess the students’ experience of both face-to-face and online case presentation assessment and to allow comparison between the two modalities. A 5-point Likert scale was used for these 9 items.

An email invitation and participant information sheet were sent to the eligible students. The first question on the MS form obtained the participant’s consent.

In addition to the Likert scale questions, 2 questions asked reasons for preference for online or face-to-face. One question asked for 3 words describing their perception of the process. Additionally, the students were asked for their perception of the most difficult part of the examination and what improvement could be made.

All 84 dental students, who had participated in the examination, were invited to take part in the study with potential participants being provided with the information regarding their participation rights and the purpose of the study, in line with the University of Glasgow policy.

### 2.4. Qualitative Analysis

The data from the questionnaires we read and re-read by the authors. This re-reading process allowed initial codings to be generated and aggregated into themes as described by Braun and Clark [16]. This process was carried out manually as described by Ryan and Bernard [17] where the texts are scrutinised and manually marked with a pen. This scrutiny facilitates themes being identified.

Alignment and consensus in the interpretation of the data were achieved through discussion by the research team.

## 3. Results

Forty-two out of eighty-four (50%) of those invited to participate completed the questionnaire.

Figure 1 summarises the responses to the nine Likert questions.

Overall, the students were confident prior to the examinations that the assessment could be carried out online. This was borne out by their perception, post-exam, where over 80% of students felt the IT had worked well.

No students felt they had insufficient information prior to sitting the examination and the majority felt that preparation for the examination was no more time-consuming than that for the face-to-face assessment.

Interaction with examiners online was split evenly with the majority either neutral or considering it easier to interact online.

Over two-thirds of the students agreed or strongly agreed that they were able to demonstrate their knowledge as well online as face-to-face. Forty percent of students were neutral as to whether the assessment was fairer online than face-to-face, but the majority found the process less stressful.

Students were split 50/50 regarding whether they would prefer to sit the examination online or face-to-face (Figure 2).

Analysis of the qualitative data revealed a number of recurring themes:-**IT/Wi-Fi;**-**Interaction with examiners;**-**Stress;**-**Environment.**

There are differences in perception of the examination between the participants who preferred to be examined online and those who preferred a live assessment.

Participants who preferred to sit the case presentation exam face-to-face.


**IT/Wi-Fi**


This aspect was particularly concerning in this sub-group: “Too much stress around IT issues. Wi-Fi issues caused stress which could have been avoided in a face-to-face environment.”


**Interaction with examiners**


Better communication and interpretation of facial expressions and body language in a face-to-face exam was cited by a number of participants as a reason for the preference for this modality:

“You can interact much better with the examiner when face-to-face and I feel communication/ body language is much clearer/ easier to gauge questions about moving on to the next part or having to expand certain answers. I feel I also have a better feeling of how I am performing during the exam when in person and it is more realistic.”


**Stress**


IT was again the major stressor for students

“Although I personally had no IT issues, the anxiety surrounding them occurring added to the overall stress of taking an exam.”


**Environment**


A small number of respondents clearly stated a preference for attending the hospital environment to undertake the examination:

“I prefer the examination setting to be in a clinical setting rather than a place at home that is meant to be relaxing.”

Participants who preferred to sit the case presentation exam online.


**IT**


Several respondents felt that the online nature of the exam allowed collation and the display of relevant case material easier.

“I felt you could see the information collated easier as it was on the screen you were looking at rather than fumbling multiple papers while stressed during the face-to-face exam”

Several respondents commented positively regarding the recording of the examination

“I also felt more confident with it being recorded that if I was unhappy with something, I could raise my concerns and the video could be reviewed.”


**Interaction with examiners**


Students perceived no difference in the interaction between online and face-to-face.

“I think the examiners were able to ask the exact same questions over Zoom than if it had been face-to-face, so I think it being online made little difference to the overall exam.”


**Stress and environment**


Within this group, there was no stress regarding IT, but there was an overlap between the themes of stress and environment but different perceptions within the themes. The ability to avoid travel and the stressful examination experience with other students around was considered an advantage.

“It was good being able to do it at home; takes away the stress of travelling in and going into the clinic, which can be a more intimidating environment.”

Table 1 summarises the differences in perception of the reported areas of difficulties overall, and for students who preferred the assessment to be face-to-face or online.

As might be expected, the students who preferred the assessment to be face-to-face were more concerned with the loss of non-verbal communication cues than those preferring online assessment. However, concern with IT issues was actually marginally higher in the students who preferred online assessment, as were the concerns with slide navigation. This is somewhat counterintuitive but perhaps reflects the more digitally literate students understanding of the technology and perception of possible errors that could occur during the examination process. Nonetheless, the percentages citing specific issues in either group were low overall.

When asked to select three words describing the assessment process, 57% of students selected fair to describe the examination. Figure 3 displays this data as a word cloud.

Most difficult issues faced during the online process

Participants were asked what they considered to be the most difficult part of sitting the online exam. Three themes emerged in response to this:IT stress/logistics;Standardization;Control of slides/PowerPoints.

## 4. Discussion

Online clinical examinations in medical sciences are not new. Various iterations have been reported in the literature since the early 2000s [18,19,20] where the assessment process evolved out of the interest in telemedicine for patient care and student education [21,22]. It was considered that this approach, utilizing new technology, would be particularly useful in remote and rural areas where travel for students or faculty to a central area for assessment would be expensive or time-consuming [23]. A recently published 12-year review of online assessment in rural and remote areas in Australia confirms the value of online assessment in these circumstances [24]. Nonetheless, prior to the pandemic, there was no widespread adoption of the online modality for assessment in medicine or dentistry. When considering the change in culture from face-to-face to remote, cost, extra examiner training, and difficulty transferring questions to a digital format were all cited as reasons for the lack of adoption of this new technology [25,26]. This changed in 2020 when COVID-19 prevented conventional clinical examination throughout the world. Many studies have been published describing the feasibility and acceptability of these remote clinical assessments: these are summarised in a recent systematic review by Kunutsor et al. [27].

Almost all papers published suggest that the change in clinical examination format from face-to-face to online is acceptable for most students. The present study is in agreement with this consensus. The majority of papers published post-March 2020 relate to the change from OSCE to VOSCE, with only one recent publication considering a case presentation examination [28]. This investigation is the first where the students can provide a direct comparison between live and virtual assessment, having completed the same assessment before and after the imposition of COVID restrictions.

Of the 84 potential respondents, 42 completed the online questionnaire, with a return rate of 50%. This is comparable to the only other similar study by Muthukrishnan et al. [28]. An average completion rate of 44.1% has been described in a recent meta-analysis of responses to online surveys, confirming that our response rate was at an acceptable level [29].

### 4.1. IT

Stress around IT logistics although IT was identified as the main issue for the group that preferred face-to-face assessment; however, it was still only a small number of students who raised it in the group. The preconception of problems around the breakdown of internet connection is common throughout the literature [11,20,24,30], but is not borne out in practice, as discussed in Kunutsor’s review [24]. In practice, fewer than 5% of students had no issues at all with IT and in no cases did IT prevent satisfactory completion of their assessment within the normal time.

The students involved in this study worked extensively online with the Zoom [14] platform in tutorials. Additionally, students rehearsed their PowerPoint presentations online with a tutor. This preparatory work carried out to ensure familiarisation for the student cohort is borne out as no students felt that they had insufficient information prior to the examination. Rehearsal has been identified as an essential part of the transition from OSCE to VOSCE [11,12,13] and was therefore adopted in this assessment.

The examiners controlled the PowerPoint presentation of the case. A few students were concerned that this prevented them from showing exactly what they wanted to discuss as easily as would have been the case in a “live” situation. Nonetheless, the students were able to ask for slides to be moved at any time during the assessment. While it is technically possible to allow the student to move the slides, there are logistical problems related to the Zoom platform if this were to be facilitated. The setup would be required every time a new student entered the virtual exam room, and this would prolong the examination process. This finding is at odds with a similar study describing VOSCE where none of the students wanted any responsibility for the movement of slides within the examination [29]. Here the difference is probably due to the student’s ‘ownership’ of the virtual poster and desire to lead the conversation with the examiner, unlike VOSCE where they do not know what the next slide may show.

### 4.2. Interaction with Examiners

The students were equivocal in their belief about whether the interaction with the examiners was the same depending on the format of the assessment. It has been suggested that VOSCEs are not suited to a purely clinical examination but are appropriate for the assessment of the student’s ability to communicate and synthesise clinical data [28]. This is precisely the gamut of skills assessed in the case presentation in our institution. Nonetheless, a minority of students felt it easier to read the non-verbal cues of the examiner in a face-to-face situation. There is potential for concealment and lack of clarity of some of the points of non-verbal communication in the online environment. It is also less easy to gauge verbal delivery and avoid interrupting individuals, as online platforms are designed for linear speech delivery. Candidates in the exam had to adapt accordingly and most felt no difficulty in this regard.

There were comments regarding the consistency of questioning and lack of standardization between cases. This was related to the examination itself rather than the medium in which it was performed. Similar comments have been observed in VOSCE examinations [30], where standardization between examiners has been suggested as a problem, but this is a function of the type of assessment rather than it being online. Nonetheless, over two-thirds of the students felt they were able to demonstrate their knowledge as well online as in person, in agreement with a previous study [28]. This is reinforced by the fact that 57% of students picked ‘fair’ as the first word to describe the online examination.

Students recognised and were reassured by the improvement in the quality assurance process within the examination.

“Easier for external examiners to jump in and out of multiple exams, easily forgot the external examiners were present so less pressure”

Movement from breakout room to breakout room within the Zoom platform allowed the external examiner to observe more examinations than was possible in previous face-to-face assessments. This concurs with oral feedback from the external examiners for this examination and previous feedback on VOSCE [11].

### 4.3. Stress

As discussed above, IT was considered one of the major causes of stress prior to the examination. However, one factor described as reducing stress was the fact that assessment on the Zoom platform would be recorded.

The recording was discussed as part of the student briefing prior to the assessment and is mentioned by Hopwood et al. [12] as useful both in the context identified by the student and for use in feedback to the candidate and training for faculty.

### 4.4. Environment

The environment in which the assessment takes place has not been explored within the dental literature. Online examination was generally seen as a necessity due to the pandemic and the importance of the examination environment was not perceived as a factor that could be varied. The split results of this study are therefore of interest. While a smaller number of students preferred the formality of the face-to-face examination in clinics, many more cited concerns around travelling, contracting Covid, and congregating with other nervous students as reasons for preferring the online format. The comfort and familiarity of their home environment were seen as much less pressurised and allowed them to feel more at ease. While a move back to the face-to-face examination will happen as the threat from COVID-19 decreases, it would be prudent to consider what changes could be made to the environment in which assessment takes place to lessen the discomfort felt by undergraduates. Whether this might be recently introduced infective strategies, such as those described by Poggia et al. [31], or continuation with safety protocol, such as described locally by governmental departments [32] or by researchers throughout the wider dental community [33].

### 4.5. Limitations of Study

A limitation of this paper is that it describes the transition from face-to-face to online assessment for a single cohort of dental students attending a single dental school in the UK. Nonetheless, this is one of the first papers examining this topic and may be useful for other schools needing to provide remote assessments.

University rules regarding the anonymity of the students taking part in any research questionnaire reduced the degree of insight into which specific students preferred either online or face-to-face assessment. It was not possible to link survey respondents to their assessment marks, allowing the analysis of whether their performance in the assessment was the main discriminator, in their preference, for online or face-to-face.

### 4.6. Future Research

The results suggest that we should look at alternative methods of assessment with more of an open mind and accept the fact that virtual assessments do work and can be more effective as they may be less stressful for the candidates and possibly staff. Further, comparative studies utilizing online and face-to-face assessment would be a welcome addition to the literature on dental assessment. While remote assessment in medicine has been ongoing for more than a decade in Australia [24], most online assessment in dentistry was a direct result of COVID-19 and consequently, there are no longitudinal data for studies of this type. Longer-term studies involving more cohorts of dental students and collaboration between different institutions would be of great value.

## 5. Conclusions

Overall, delivery of the case presentation assessment is possible online.

Online case presentation is a fair, useful, and authentic assessment appropriate to the needs of faculty and students.

The students’ satisfaction with the two possible methods of conducting this assessment suggests it would be reasonable to conduct this examination online in the future.

Covid has been a catalyst for change within tertiary education. While pandemic restrictions have reduced throughout the world and the necessity for online adaptation to assessment in dentistry has receded, there is no reason why assessments of this type should not be conducted in the virtual environment. Furthermore, this successful adaptation to online assessment, as has been the case with OSCE examinations, provides a degree of future-proofing against any further resurgence of the pandemic, or other emergency situations.

## Figures and Tables

**Figure 1 dentistry-11-00045-f001:**
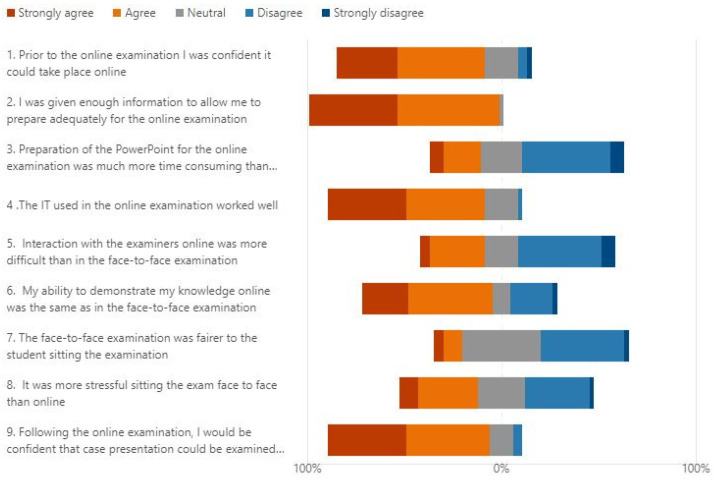
Participant responses regarding the virtual case presentation exam.

**Figure 2 dentistry-11-00045-f002:**
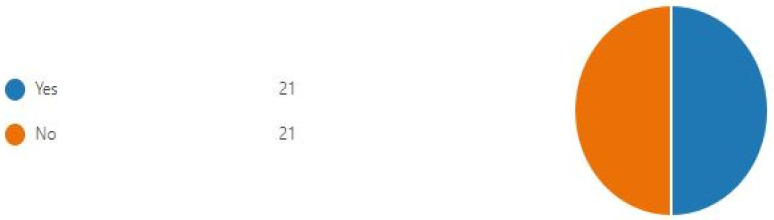
Pie chart showing whether participants would prefer to sit the case presentation exam online.

**Figure 3 dentistry-11-00045-f003:**
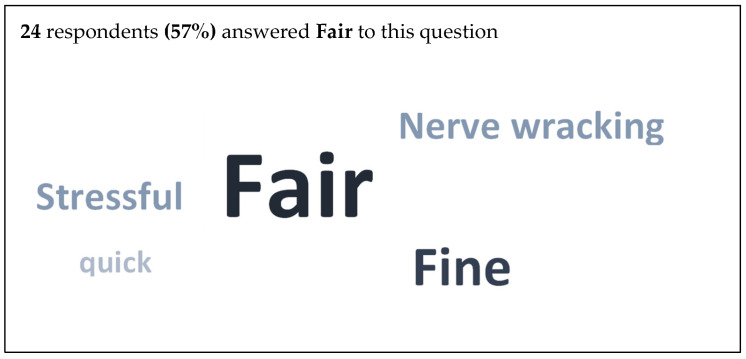
Word cloud: participants were asked for three words to describe the online examination experience.

**Table 1 dentistry-11-00045-t001:** Summary of differences in perception between students who preferred online or face-to-face assessment.

Most Reported Issues	Overall	Preferred Face-to-Face	Preferred Online
IT concerns	36%	33%	38%
Navigating slides	14%	9%	19%
Non-verbal communication	5%	9%	0%
Exam Stress	12%	9%	14%

## Data Availability

Research data are not shared due to the requirements of the ethical approval granted.

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
