# Peer review of "Adaptation to Virtual Assessment during the COVID-19 Pandemic: Clinical Case Presentation Examination"

_dentistry, 2023, doi:10.3390/dj11020045_

Round 1

Reviewer 1 Report

The manuscript is interesting, current if we consider that we still have effects after the covid-19 pandemic.

However, some important issues must be evaluated by the authors:

- Throughout the text, all abbreviations must be properly identified, in full, when used for the first time.

Around the world the most serious period of the covid-19 pandemic was really challenging for teachers and students. Undoubtedly, there has been and will be some kind of impairment in learning, especially for areas where practical clinical training is essential, such as dentistry.

Most of us teachers were forced to make a rapid transition to online media for which we were not properly prepared. Likewise, this happens for our students.

It is evident, yes, that a lot was learned by everyone and that the use of online means can help in an important way and bring great teaching-learning experiences.

However, in my view, and that of several of my colleagues, face-to-face contact between student and teacher is essential and should not be diminished.

One of the biggest concerns we had was to maintain the seriousness of the evaluation processes, which demands greater preparation of the tests. ensuring that the student does not have access to other means on which to rely.

How to guarantee this? How to have real control that the student did not use other means to take the applied exams?

Materials and Methods

Well described

Results

Only 42 out of 84 students responded to the questionnaire. This does not seem satisfactory to me, denoting one of the problems of using online media, which is the effective participation of the whole group.

In a face-to-face assessment, everyone would be present, no doubt.

Thus, I disagree when the authors say that the participation obtained is satisfactory.

One sentence on p. 6, highlighted by the authors, caught my attention: “It was good being able to do it at home- takes away the stress of traveling in and going into the clinic, which can be a more intimidating environment.” See, they will have to see their patients in the clinic, so one would think that this would be intimidating for them? That's exactly what they're being prepared for. I understand that online activity, in this respect, is more problematic than good, as it creates this distortion. of thought.

One of the biggest concerns we've been feeling is the desire of large private trial groups trying to get authorization for their courses to be based on distance education, all based on "great online experiences". It is a matter of concern for areas such as Dentistry and Medicine, in which direct contact with the patient is essential for carrying out a large part of our activities. This should be considered in this article in depth.

Author Response

The abbreviations have been checked and described in full at their first use within the text

This exam was a live online examination. The student was on the screen at all times during the examination. It was not like an essay or MCQ examination where they could, potentially, access other resources to help them perform better in the assessment. They were either talking or being asked questions during the whole process and were visible to two internal examiners and an external examiner. They would have to access other resources in real time while being watched to do what the referee suggests. We haver described this aspect of the examination process more fully in the text.

While we agree with the reviewer that face to face , in the same room, is the best way to conduct assessment. This was not possible during the pandemic and what we used was, in our opinion, the best possible substitute. It was the same exam, conducted in the same way, but remotely.

To clarify the referees point regarding participation. We had 100% participation in the examination. There was no difference between real and virtual, in this regard. The completion rate for the questionnaire post examination was 50%. While the referee may not like the response rate a recent meta analysis (below) shows that our response rate in very similar to other studies in the field and as such is acceptable. This was already stated in the paper.

  1. Wu M-J, Zhao K, Fils-Aime F. Response rates of online surveys in published research: A meta-analysis. Computers in Human Behaviours Report. 2022; 7: https://doi.org/10.1016/j.chbr.2022.100206

It would not be possible, within our institution, to impose the completion of any questionnaire; whether online or in person. Any participation in such research activities are voluntary on the  part of the student. Ethics would not allow any coercion to take part.

The reviewer stated - One sentence on p. 6, highlighted by the authors, caught my attention: “It was good being able to do it at home- takes away the stress of traveling in and going into the clinic, which can be a more intimidating environment.” 

It is interesting that on page 5 another student takes the opposite view. At the time of the examination there was particular stress around travel, particularly the use of public transport and contracting Covid. It is likely that this is part of what the student is alluding to in the stressful nature of travelling to the dental school. While stress is indeed part of the clinical environment and clinical assessments do indeed require direct patient contact, certain aspects of assessment, such as that described in this submission (essentially presentation of a treatment plan) are entirely amenable to the online format.

While the authors concur with the referees concerns regarding authorisation of large online courses and education at a distance we do not feel this is necessarily relevant for this single individual assessment as long as other face to face methods of assessing students in clinics are being used (which they are!). This is not the basis of the paper, not what we were investigating, and we do not feel it is within the scope of this investigation to discuss this in depth, as suggested.

Reviewer 2 Report

This is a high-quality manuscript with a smooth narrative and a succinct but thorough description of the results, and it also features explanatory graphics.

However, there is an acronym (MSA) that needs elaboration (pg 1, line 42)

Also, please insert a comma (,) on page 2, line 82.

Author Response

Thanks to the reviewer for his comments. The acronym and comma mentioned have been corrected as suggested.

Reviewer 3 Report

Thank you for the opportunity to review the manuscript “Adaptation to virtual assessment during the Covid-19 pandemic: Clinical Case Presentation Examination”. The authors evaluate student’s impression of an online Case Presentation assessment during the Covid-19 pandemic and compare it with a face-to-face examination. Study results are gathered using an online, self-administered survey. However, there is a lack of more detailed information and in-depth analysis of what lies behind the students' answers, which would strengthen the conclusions drawn.

My comments/questions:

1.       What do the abbreviations MSA (line 42), VOSCE (line 263) mean?

2.       What was the time frame between the online examination and the invitation to participate (participation) in the survey? It is important to know how “fresh” the students’ experience is.

3.       Were all the students invited to take part in the survey, regardless of their grade in the exam? Have you noticed if the exam grade affected students’ answers? Have you noticed if previous experience on online (or face-to-face) case presentation (not necessarily during the exam) affected students’ answers?

4.       Forty-two students participated in the survey. Could you please describe your study sample in more detail (e.g., distribution of students according to the grade received, the complexity of the case presented)?

5.       Was the face-to-face case presentation before the pandemic recorded? I would suggest comparing and discussing the issue of the objectivity of the online and face-to-face examination.

6.       What was the percentage distribution of the most difficult issues faced during online process? Could you also indicate in the results section the percentage distribution of the most important issues (i.e., IT/Wifi, Interaction with examiners, Stress, Environment) in each group (i.e., participants who preferred to sit the case presentation exam face to face and online)?

7.       Conclusions should not repeat the results.

8.       What changes could be made to the online exam in the light of students’ responses and recommendations?

Author Response

Reviewer 3 comments and answers

  1. What do the abbreviations MSA (line 42), VOSCE (line 263) mean?

MSA is addressed on line 42

VOSCE is mentioned previously in the text on line 61

  1. What was the time frame between the online examination and the invitation to participate (participation) in the survey? It is important to know how “fresh” the students’ experience is.

The time between the online examination and the invitation to participate was longer than we would have liked due to university policy at that time. Ethical approval during the height of Covid took months to process and we were advised that no surveying post assessment should take place for at least 3 months. In the end it was around a year between examination and the results being obtained

  1. Were all the students invited to take part in the survey, regardless of their grade in the exam? Have you noticed if the exam grade affected students’ answers? Have you noticed if previous experience on online (or face-to-face) case presentation (not necessarily during the exam) affected students’ answers?

  1. Forty-two students participated in the survey. Could you please describe your study sample in more detail (e.g., distribution of students according to the grade received, the complexity of the case presented)?

Q3+4 All the students were invited to take part in the survey. The survey was anonymous, in line with university policy. Consequently. it was not possible to assess if the students grade affected their answers the proportion of respondents related to the grade they received or the complexity of their case.

  1. Was the face-to-face case presentation before the pandemic recorded? I would suggest comparing and discussing the issue of the objectivity of the online and face-to-face examination.

Unfortunately, no and mock examinations are not routinely recorded. In retrospect it would have been helpful. We weren’t expecting a pandemic.

  1. What was the percentage distribution of the most difficult issues faced during online process? Could you also indicate in the results section the percentage distribution of the most important issues (i.e., IT/Wifi, Interaction with examiners, Stress, Environment) in each group (i.e., participants who preferred to sit the case presentation exam face to face and online)?

A table has been produced and entered in the paper in answer to your question. Overall, there was little difference between the two groups in this regard. The numbers are too small to carry out meaningful statistical analysis.

  1. Conclusions should not repeat the results.

The conclusion has been adjusted to make it more succinct.

  1. What changes could be made to the online exam in the light of students’ responses and recommendations?

The examination seemed to work well. The equivocal nature of the students’ responses suggests that there is no specific change that will be universally deemed positive by all. It would be preferable for students to be able to move the slides themselves within the breakout room. Apart from that there was very little difference from the live examination. The virtual examination has actually informed our return to live examinations where less, rather than more information (such as study models, extra photographs) are brought to the exam. These were always a source of worry for the students but often proved superfluous during the assessment. A slimming down of requirements has proved easier for staff and students alike. I will add a little of this to the discussion.

Reviewer 4 Report

The authors performed a study in a very interesting topic. As a director of the OSCE examination the last 5 years, in which an online model was never applied (even during COVID) to the 2nd year predoctoral dental students, I have some points-questions that should be addressed in the materials and methods and discussion sections. 

- Which OSCE stations were used at the online vs the in person exam? Please describe the difficulties to change a clinical exam to a non-clinical exam and give examples. Specifically, in periodontics, the predoctoral students need to identify and demonstrate on a typodont, the use of specific instruments like the 4R/4L, periodontal probe, Nabers probe etc. How you can tranfer that question and evaluate their dexterity and knowledge on an online exam? 

- My opinion is that the structure of the paper should change from a possible online exam conclusion to experience and difficulties of a clinical exam to online due to pandemic. These specific clinical exams simply can't be online, the have to be in person on a clinical setting. Pandemic never stopped any School of Dentistry in the USA to perform OSCE, why that was a problem in the UK? 

Author Response

The paper does not deal with an OSCE examination, it was a case presentation examination. The authors have published 2 papers regarding OSCE examinations on line (VOSCE) and would draw the referees attention to the excellent review paper by Knutford et al.

  1. Kunutsor SK, Metcalf E, Westacott R, Revell LJ, Blythe AJ. Are Remote Clinical Assessments a Feasible and Acceptable Method of Assessment? A Systematic Review. Medical Teacher. 2021; https://doi.org/10.1080/0142159X.2021.1987403

While we would agree that specific practical skills, as described by the referee are not possible by Virtual OSCE; many of the skills tested in OSCE are. In Scotland we have moved away from single simulated tests of students' manual skills and prefer to use our longitudinal clinical assessment tools as these give a more accurate and defensible assessment more in line with our regulators preference.

This was a case presentation examination where the ability of the student to bring together clinical information they had assimilated live in clinic and present it in a clear and concise manner, answer clinical questions related to their case was being assessed. There were no manual skills being examined.

Round 2

Reviewer 1 Report

The corrections made to the manuscript made it better.

All queries have been answered.

The pre

Author Response

Thank you for your response.

Reviewer 3 Report

Thank you for answering the questions. I have no further comments.

Author Response

Thank you for your response

Reviewer 4 Report

I would like to thank the authors for their corrections/answers/comments. However, I would like to address some points that should be clearly addressed and discussed in the paper; 

1. OSCE can't be replaced by VOSCE

2. COVID was a different situation, an exception, which some schools NOT in the US, replaced it with another exam. All the schools in USA during the pandemic, with the use of proper PPE followed their curriculum including in person OSCE examination. 

3. The systematic review you are referring to was based on studies with low methodological quality and small sample sizes. In addition, in their discussion section, they state that VOSCE have several limitations such as not being able to assess clinical examination skills or practical skills. 

4. Clinical case presentation is an additional acceptable way to assess clinical reasoning competences and communication skills but it can't replace OSCE. 
